# Prognostic value of pre-treatment systemic immune-inflammation index in patients with endometrial cancer

**Sho Matsubara**, **Seiji Mabuchi***, **Yoshinori Takeda**, **Naoki Kawahara**, **Hiroshi Kobayashi**

Department of Obstetrics and Gynecology, Nara Medical University, Kashihara, Nara, Japan

* smabuchi@naramed-u.ac.jp

## Abstract

### Background

The systemic immune-inflammation index (SII), which is calculated using absolute platelet, neutrophil, and lymphocyte counts, has recently attracted attentions as a prognostic indicator in patients with solid malignancies. In the current study, we retrospectively investigated the prognostic significance of pre-treatment SII among patients with endometrial cancer.

### Method

Endometrial cancer patients treated at Nara medical university hospital between 2008 and 2018 were included in the current study. Receiver operating characteristic (ROC) curve was used to find the optimal SII cut-off values for 3-years progression free survival (PFS) and overall survival (OS). Then, the predictive abilities of SII and its superiority over neutrophil to lymphocyte ratio (NLR) and platelet to lymphocyte ratio (PLR) were investigated. Kaplan-Meier method was used to calculate the OS and PFS rates, and log-rank test was used to compare the survival rate between two groups. Univariate and multivariate Cox regression analysis were performed to identify risk factors for PFS and OS.

### Result

A total of 442 patients were included in the current study. The cut-off value of SII for predicting PFS and OS were defined by ROC analysis as 931 and 910, respectively. Univariate analyses showed that elevated SII was associated with significantly shorter survival ($p$ <0.001 for both PFS and OS). Cox regression analyses revealed that an advanced FIGO stage ($p$ <0.001 for both PFS and OS) and an elevated SII ($p$ = 0.014 for PFS, $p$ = 0.011 for OS) are the independent prognostic factors for survival. When SII was compared with NLR and PLR, SII showed greater area under curve for predicting survival.

### Conclusion

The SII is an independent prognostic factor in endometrial cancer patients, allowing more precise survival estimation than PLR or NLR.

**Data Availability Statement:** There is an ethical restriction on sharing a de-identified data set, because the data contain potentially identifying patient information. However, data from this study are available upon request. Ethics Committee of

Nara Medical University has imposed them. Contact information for a data requests is described bellow. Department of Obstetrics and Gynecology, Nara Medical University. Address: 840 Shijo-cho, Kashihara, Nara, 634-8522 Japan. Telephone: +81-744-29-8877 FAX: +81-744-23-6557 E-mail: obgynlab@naramed-u.ac.jp.

**Funding:** The authors received no specific funding for this work. Thus, no funders had role in study design, data collection and analysis, decision to publish, or preparation of the manuscript.

**Competing interests:** There are no conflicts of interest to declare.

## Introduction

Endometrial cancer is the second most common and the fourth leading cause of death due to gynecological cancer among women worldwide, and there are an estimated 382,069 new cases and 89,929 deaths attributed to endometrial cancer worldwide in 2018 [1]. In Japan, estimated 11,120 new cases and 2526 deaths were reported annually in 2017 [2,3]. The 5-year survival rate is 74–91%, 57%-66%, and 20–25% for patients with FIGO stage I-II, stage III, and stage IV disease, respectively [4].

Numerous studies have attempted to identify prognostic factors in patients with endometrial cancer. As a result, International Federation of Gynecologic and Obstetrics (FIGO) stage, the histologic subtypes and the grade of the tumor, positive peritoneal cytology and lymphovascular space invasion (LVSI) have been pointed out as prognostic factors [5–11]. However, most of these prognosticators can be available only after surgical treatment, and the ability of these risk factors to predict recurrence and estimate survival has been insufficient.

Inflammatory reactions in tumor microenvironment plays an important role in tumor development and progression [12,13]. As inflammation can stimulate granulopoiesis or thrombopoiesis, inflammatory indexes such as leukocytosis, neutrophilia, thrombocytosis, neutrophil to lymphocyte ratio (NLR), and platelet to lymphocyte ratio (PLR) can serve as significant prognosticator in patients with solid malignancies including endometrial cancer [14,15]. Recently, a novel inflammatory index, the systemic immune-inflammation index (SII) based on peripheral neutrophil, platelet, and lymphocyte counts, was been found to be an useful prognosticator in cancer patients [16–30]. However, the prognostic significance of SII in endometrial cancer patients or its relative utility when compared with other inflammatory indexes has not been fully investigated [31,32]. In the current study, using clinical data obtained from 442 endometrial cancer patients, we investigated the prognostic significance of SII in endometrial cancer.

## Material and method

### Patients

This retrospective cohort study was approved by the Ethics Committee of Nara Medical University, and the analysis of the patient-derived data was carried out in accordance with the Declaration of Helsinki.

We generated a database of patients who were diagnosed with endometrial cancer from January 2008 to December 2018 at Nara medical university hospital. All patients signed the informed consent form. All patients routinely took blood examination before the initiation of treatment. Clinical information including age, body mass index (BMI), histological subtype, histological grade, FIGO stage, LVSI, peritoneal cytology, and pre-treatment blood examination results were collected from the database, and retrospectively reviewed.

### Treatment and follow-up

Patients were staged according to the FIGO surgical staging criteria. If surgery was not possible especially in stage IV patients, patients were clinically staged instead. Almost all patients except for those with tumor invasion to pelvic organs or with systemic metastases (FIGO stage IV) were treated with surgery followed by adjuvant chemotherapy. Some patients who cannot be treated with chemotherapy because of poor general condition were treated only with surgery. The main surgical procedures consisted of total abdominal hysterectomy (TAH) or total laparoscopic hysterectomy (TLH), bilateral salpingo-oophorectomy (BSO), and pelvic lymphadenectomy. Patients who exhibited grade 3 endometrioid adenocarcinoma, tumor invasion

more than 50% of the myometrium, non-endometrioid histology or extrauterine disease, para-aortic lymphadenectomy was also performed. In case with cervical stromal invasion, radical hysterectomy was employed instead of TAH/TLH. In the case of non-endometrioid histology, omentectomy was also performed. Postoperative adjuvant chemotherapy or radiotherapy was recommended for all patients who had any of the following risk factors for recurrence: over 50% myometrial invasion, cervical stromal invasion, extra-uterine disease, and high-risk histological type and grade (non-endometrioid, grade 3 endometrioid). The adjuvant chemotherapy includes triweekly cisplatin (50 mg/m$^2$) plus doxorubicin (60 mg/m$^2$) or triweekly paclitaxel (175 mg/m$^2$) plus carboplatin (area under the curve: 5). When the patients had tumors invaded to pelvic organs or had systemic distant metastases (FIGO stage IV), personalized treatment either with chemotherapy, radiotherapy, or best supportive care were performed.

Follow-up was conducted one to three monthly for the first 36 months, 6-monthly for the next 24 months in outpatient clinic. Lesions suspected for recurrence were confirmed by histological or cytological diagnosis, whenever possible. In cases in which the histology or cytology of the recurrent lesion could not be assessed, the diagnosis of recurrence was made based on imaging studies.

## Definitions of systemic inflammatory indexes and statistical analysis

The inflammatory indexes were calculated preoperative inflammatory indicators with the following formulas; SII = platelet counts × neutrophil counts/lymphocyte counts, NLR = neutrophil counts/lymphocyte counts, PLR = platelet counts/lymphocyte counts. In cases chemotherapy had been employed as a primary treatment, SII was calculated from the pretreatment inflammatory indicators.

Using cancer-specific death, cancer recurrence, and disease progression in 3-years as the end point, receiver operating characteristic (ROC) curve analysis was performed to determine the cut-off value of SII. The cut-off value was based on the highest Youden index (i.e., sensitivity + specificity– 1). ROC curve analysis was also used to compare the usefulness of the three inflammatory indexes SII, NLR, PLR as prognostic factors. Pearson chi-square test was used to compare between groups. Kaplan-Meier method was used to calculate the overall survival (OS) and progression free survival (PFS) rates, and log-rank test was used to compare the survival rate between two groups. OS was defined as the time from the initiation of treatment to death of any cause, and PFS was defined as the time from the initiation of treatment until evidence of disease progression or death of any cause. Univariate and multivariate survival analysis were performed in order to identify independent risk factors for prognosis by using Cox regression analysis. Differences with $P$-values less than 0.05 were considered statistically significant. SPSS software (ver.25.0, SPSS Inc., Chicago, IL) and R software (ver.3.6.2, freely available from URL: https://www.r-project.org/) were used for the statistical analysis.

## Result

### Patients

A total of 442 endometrial cancer patients were included in the present study (Table 1). The median age was 59 years (range: 24–84). The median follow-up time was 47.5 months (range: 2–138). Three hundred and eleven patients (70.4%) had stage I, 38 (8.5%) had stage II, 56 (12.6%) had stage III and 37 (8.3%) had stage IV disease. Ninety six patients (21.7%) had non-endometrioid type histology, and 151 patients exhibited high histological grade.

ROC analyses revealed that the optimal cut-off values of SII for predicting PFS and OS were 931 and 910, respectively (Fig 1). As shown in Table 2, an elevated SII (>931) was observed in

**Table 1. Clinicopathological characteristics of patients with endometrial cancer.**

| | | No. of Patients (n = 442) | % |
|---|---|---|---|
| Age | Mean ± SD (range) | 59.0 ± 11.5 (24–84) | |
| | ≦59 | 233 | 52.7 |
| | >59 | 209 | 47.3 |
| BMI | Mean ± SD (range) | 23.3 ± 5.4 (15.1–47.8) | |
| FIGO stage | I | 311 | 70.4 |
| | II | 38 | 8.5 |
| | III | 56 | 12.6 |
| | IV | 37 | 8.3 |
| Histological subtype | Endometrioid | 346 | 78.3 |
| | Non-endometrioid | 96 | 21.7 |
| Histological grade | Grade 1 | 198 | 44.8 |
| | Grade 2 | 93 | 21.0 |
| | Grade 3 [1] | 151 | 34.2 |
| LVSI | Positive | 131 | 29.6 |
| | Negative | 278 | 65.2 |
| | NA | 13 | 2.9 |
| Peritoneal cytology | Positive | 81 | 18.3 |
| | Negative | 328 | 78.7 |
| | NA | 13 | 2.9 |
| Primary treatment | Surgery alone | 201 | 45.5 |
| | Surgery + adjuvant chemotherapy | 218 | 49.3 |
| | Surgery + adjuvant radiotherapy | 22 | 5.0 |
| | Chemotherapy alone | 1 | 0.2 |
| | Radiotherapy alone | 0 | 0 |

[1]: Grade 3 includes grade 3 endometrioid type and non-endometrioid type.

SD; standard deviation, BMI; body mass index, FIGO; Federation of Gynecology and Obstetrics, LVSI; lymphovascular space invasion, SII; systemic immune-inflammation index, NA; not available.

135 patients (30.5%), and was associated with advanced FIGO stage, non-endometrioid histology, higher tumor grade, LVSI, and positive peritoneal cytology. Similar results were obtained from the analysis in which a cut-off value was 910 was employed (S1 Table).

## Correlation between the SII and the prognosis

During follow-up, cancer recurrence was observed in 74 patients (16.7%), 58 patients (13.1%) died of endometrial cancer and 2 patients died of other diseases. As shown in Fig 2, patients exhibiting higher SII showed significantly shorter PFS and OS than those exhibiting lower SII. Cox regression analyses revealed that an advanced FIGO stage (hazard ratio: 5.01; 95% CI: 3.06 to 8.19; $p <0.001$ for PFS, hazard ratio: 9.08; 95% CI: 4.83 to 17.03; $p <0.001$ for OS) and an elevated SII (hazard ratio: 1.71; 95% CI: 1.11 to 2.62; $p = 0.014$ for PFS, hazard ratio: 1.96; 95% CI: 1.16 to 3.30; $p = 0.011$ for OS) were the independent prognostic factors for endometrial cancer patients for both PFS and OS (Table 3).

## Comparison of SII with other systemic inflammatory indexes

Finally, using ROC curves, we compared the ability of various systemic inflammatory indexes to predict PFS and OS (Fig 3). As shown, although there is no significant statistical deference,

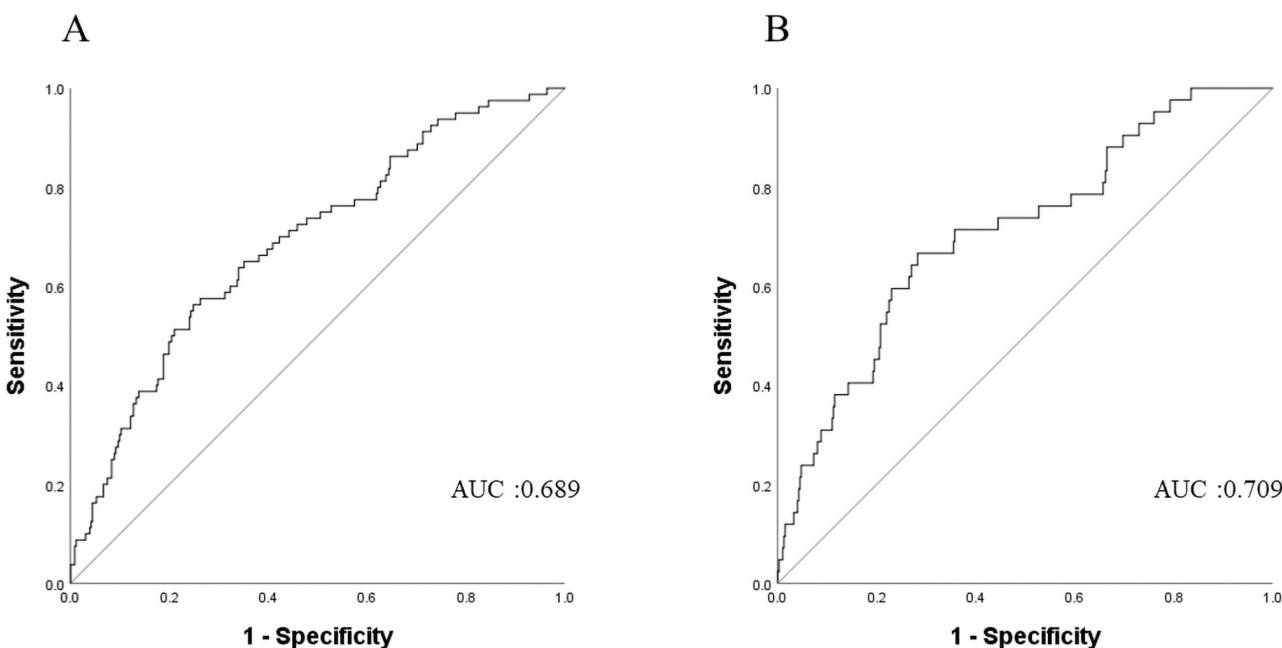

**Fig 1. Clinical implications of SII in endometrial cancer patients.** A. ROC curves for progression-free survival (PFS) at 3-years. B, ROC curves for overall survival (OS) at 3-years.

**Table 2. Correlations between SII for PFS and clinicopathological characteristics.**

| | | SII≦931 (n = 307) | SII>931 (n = 135) | *p*-value |
|---|---|---|---|---|
| Age | ≦59 | 156 | 77 | 0.228 |
| | >59 | 151 | 58 | |
| FIGO stage | I | 239 | 72 | <0.001 |
| | II | 26 | 12 | |
| | III | 30 | 26 | |
| | IV | 12 | 25 | |
| Histological subtype | Endometrioid | 249 | 97 | 0.029 |
| | Non-endometrioid | 58 | 38 | |
| Histological grade | Grade 1 | 159 | 41 | <0.001 |
| | Grade 2 | 62 | 31 | |
| | Grade 3 [1] | 86 | 63 | |
| LVSI | Positive | 84 | 47 | 0.001 |
| | Negative | 219 | 79 | |
| | NA | 4 | 9 | |
| Peritoneal cytology | Positive | 47 | 34 | <0.001 |
| | Negative | 256 | 92 | |
| | NA | 4 | 9 | |

[1]: Grade 3 includes grade 3 endometrioid type and non-endometrioid type.

PFS; progression free survival, SII; systemic immune inflammation index, FIGO; Federation of Gynecology and Obstetrics, LVSI; lymphovascular space invasion, NA; not available.

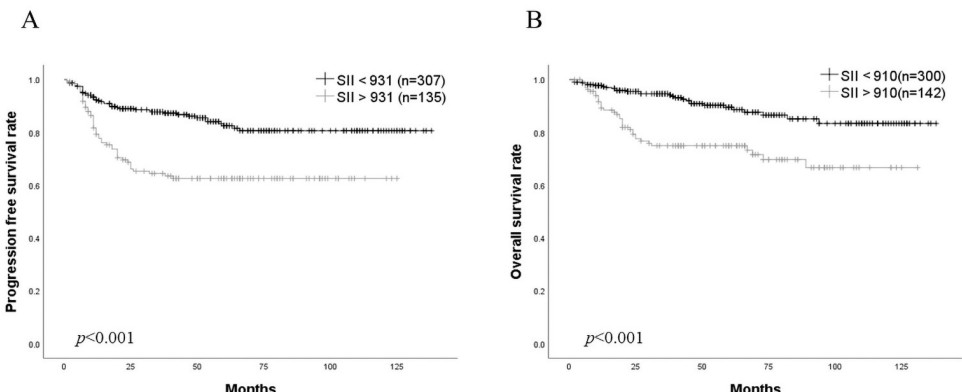

**Fig 2. Kaplan-Meier estimates of survival in endometrial cancer patients based on SII.** A, PFS. B, OS. Patients exhibiting higher SII showed significantly shorter PFS (p<0.001) and OS (p<0.001) than those exhibiting lower SII. The p-values were analyzed by log-rank test.

SII showed larger AUC than NLR. Moreover, SII showed significantly larger AUC than PLR, indicating the superiority of SII when compare with other inflammatory indexes in this patient population (Table 4).

## Discussion

To our knowledge, this is the largest study that investigated the prognostic significance of SII in endometrial cancer patients (S2 Table). In our study population, when cut-off values of 931 and 910 were employed, elevated SII was observed in 30.5% and 32.1% of patients, respectively. An elevated SII was significantly associated with an advanced clinical stage, non-endometrioid

**Table 3. Result of univariate and multivariate cox regression analysis for progression free survival and overall survival in endometrial cancer.**

| Variables | | PFS | | | | OS | | | |
|---|---|---|---|---|---|---|---|---|---|
| | | Univariate analysis | | Multivariate analysis | | Univariate analysis | | Multivariate analysis | |
| | | HR (95% CI) | *p*-value | HR (95% CI) | *p*-value | HR (95% CI) | *p*-value | HR (95% CI) | *p*-value |
| FIGO stage | Stage I, II | 1 | | 1 | | 1 | | 1 | |
| | Stage III, IV | 8.85 (5.79–13.53) | <0.001 | 5.01 (3.06–8.19) | <0.001 | 8.85 (5.78–13.53) | <0.001 | 9.08 (4.83–17.03) | <0.001 |
| Histological subtype | Endometrioid | 1 | | 1 | | 1 | | 1 | |
| | Non-endometrioid | 3.73 (2.48–5.63) | <0.001 | 1.97 (1.08–3.57) | 0.026 | 3.73 (2.48–5.63) | <0.001 | 1.87 (0.93–3.75) | 0.079 |
| Histological grade | Grade 1, 2 | 1 | | 1 | | 1 | | 1 | |
| | Grade 3 [1] | 4.26 (2.78–6.52) | <0.001 | 1.57 (0.84–2.91) | 0.156 | 4.23 (2.78–6.52) | <0.001 | 2.18 (1.03–4.62) | 0.043 |
| LVSI | Negative | 1 | | 1 | | 1 | | 1 | |
| | Positive | 2.79 (1.85–4.20) | <0.001 | 1.45 (0.92–2.30) | 0.108 | 2.79 (1.85–4.20) | <0.001 | 1.17 (0.67–2.01) | 0.584 |
| Peritoneal cytology | Negative | 1 | | 1 | | 1 | | 1 | |
| | Positive | 2.64 (1.71–4.07) | <0.001 | 1.18 (0.74–1.87) | 0.484 | 2.64 (1.71–4.07) | <0.001 | 0.82 (0.54–1.63) | 0.824 |
| SII | ≤931 | 1 | | 1 | | | | | |
| | >931 | 2.58 (1.71–3.88) | <0.001 | 1.71 (1.11–2.62) | 0.014 | | | | |
| | ≤910 | | | | | 1 | | 1 | |
| | >910 | | | | | 2.48 (1.65–3.74) | <0.001 | 1.96 (1.16–3.30) | 0.011 |

[1]: Grade 3 includes grade 3 endometrioid type and non-endometrioid type.

FIGO; Federation of Gynecology and Obstetrics, LVSI; lymphovascular space invasion, SII; systemic immune inflammation index, NA; not available, PFS; progression free survival, OS; overall survival, HR; hazard ratio, CI; confidence interval.

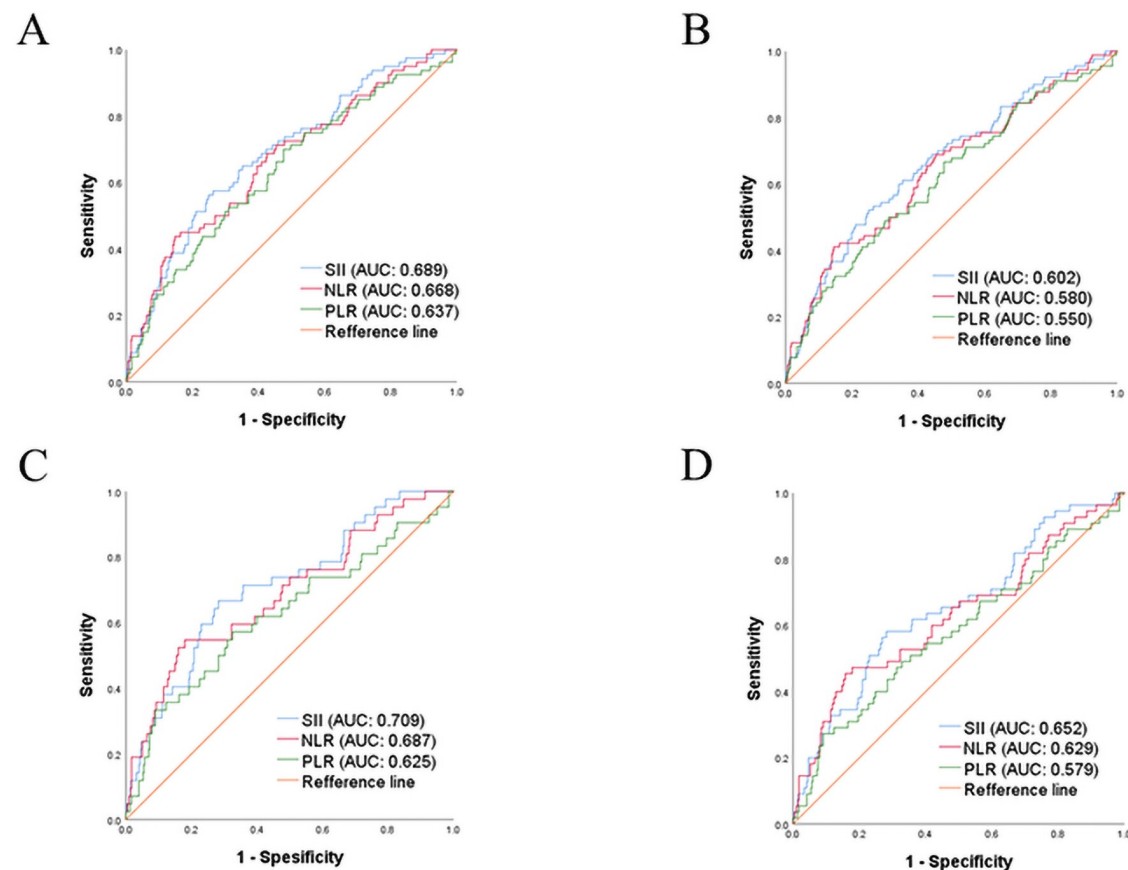

**Fig 3. Comparison of the utility of the SII versus theta of the other inflammatory indexes.** A, ROC curves for 3-years PFS. B, ROC curves for 5-years PFS. C, ROC curves for 3-years OS. D, ROC curves for 5-years OS.

histological subtype, higher histological grade, LVSI and positive peritoneal cytology, indicating the aggressive nature of endometrial cancer displaying elevated SII. In the survival analysis, SII was found to be a significant predictor for decreased PFS and OS in patients with endometrial cancer. We also found that the SII was a superior prognostic predictor to NLR and PLR in patients with endometrial cancer.

**Table 4. Correlations between SII and other systemic inflammatory indexes.**

|  |  | OS | | PFS | |
|---|---|---|---|---|---|
|  |  | AUC | *p*-value (versus SII) | AUC | *p*-value (versus SII) |
| 3-years | SII | 0.709 |  | 0.689 |  |
|  | NLR | 0.687 | 0.365 | 0.668 | 0.208 |
|  | PLR | 0.625 | 0.008 | 0.637 | 0.041 |
| 5-years | SII | 0.652 |  | 0.602 |  |
|  | NLR | 0.629 | 0.268 | 0.580 | 0.197 |
|  | PLR | 0.579 | 0.022 | 0.550 | 0.041 |

SII; systemic immune-inflammation index, OS; overall survival, PFS; progression free survival, AUC; area under curve, NLR; neutrophil-lymphocyte ratio, PLR; platelet-lymphocyte ratio.

In endometrial cancer, 2 previous studies have investigated the prognostic significance of SII. As shown (S2 Table), both studies included relatively small number of patients, but produced conclusions were similar to ours. The consistent results obtained from these studies strongly indicate the prognostic significance of SII in endometrial cancer patients. However, as the cutoff value of SII differed between the studies, further multi-institutional investigations including larger number of patients are necessary.

The result obtained in the current study may have important clinical implication. By performing simple and low-cost peripheral blood examinations, we may be able to identify patients who have greater risk for developing recurrence. In our study population, elevated SII was detected in roughly 30% of patients. For these patients, careful post-treatment follow-up can be performed. Moreover, the results of our ROC analysis suggest that the prediction of recurrence can be significantly improved by assessing SII instead of PLR or NLR, allowing more precise survival estimation than PLR or NLR.

The underlying mechanisms responsible for the development of elevated SII in endometrial cancer patients and the subsequent increase in the aggressiveness of the disease remain unknown. However, previous studies have suggested that lymphocytes interplay in controlling tumor growth via secreting cytokines such as interferon gamma (IFN-γ) and tumor necrosis factor alpha (TNF-α). Thus, low lymphocyte counts may reflect the impaired host immunosurveillance, which might lead to a poor prognosis [33,34]. Moreover, as an elevated SII reflects a status of elevated neutrophil and platelet counts, we believe that factors that stimulate granulopoiesis and/or thrombopoiesis might also be involved in the mechanism responsible for the elevated SII in endometrial cancer patients. According to a recent study conducted by Yokoi et al, the tumor derived granulocyte colony-stimulating factor (G-CSF) induces leukocytosis in endometrial cancer patients and stimulates the production of interleukin-6 (IL-6) from tumor microenvironment and cause thrombocytosis [35]. As both neutrophils and platelets can stimulate cancer progression by enhancing tumor angiogenesis, tumor cells' epithelial mesenchymal transition (EMT) via the production of vascular endothelial growth factor (VEGF), matrix metalloproteinase 9 (MMP-9) and IL-6 or IL-8, it is possible that tumor become aggressive in patients with neutrophilia and thrombocytosis. [36–39]. We consider that future investigation of the underlying causative mechanism of elevated SII will aid the development of novel effective treatments for this type of endometrial cancer.

There are some limitations of this study. First, our study was conducted at a single institution. Second is the retrospective nature of this study, which is susceptible to bias in data selection and analysis. Third, we included only Japanese women with endometrial cancer in which increased BMI is rarely observed (mean BMI of the patients was 23.3). Thus, further investigations involving patients with wide range of BMI will be required. Fourth, although SII was found to be an independent predictor for decreased PFS and OS in patients with endometrial cancer in the multivariate analysis, the prognostic significance of SII was not investigated in studies in which arms (SII-high versus SII-low) were balanced. Fifth, we did not include comorbid conditions such as a diabetes or cardiovascular diseases, which may affect patient's survival, as a prognostic variable in our univariate and multivariate analyses. Sixth, as this study is covering a long study period, changes in the choice of treatments might have affected the survival of patients included in the study. To eliminate these biases and validate the results from current study, future larger studies especially in a prospective randomized or a propensity score matching settings are required. Finally, in our multivariate analyses, the prognostic impact of FIGO stage was greater than SII (Table 3). However, whether it is reasonable to apply SII over FIGO stage or vice versa to estimate the prognosis of endometrial cancer patients remains unknown. Thus, the clinical application of SII in the survival estimation need to be further investigated in this patient population.

In conclusion, the presence of elevated SII at the time of initial diagnosis is an independent predictor of poor prognosis in endometrial cancer patients. Future validation study, especially in large scale prospective setting, is warranted so that SII will be widely used as a prognosticator in endometrial cancer patients.

## Supporting information

**S1 Table. Correlations between SII for OS and clinicopathological characteristics.**
(DOCX)

**S2 Table. Summary of investigations of the role of SII in endometrial cancer patients.**
(DOCX)

## Acknowledgments

The authors thank Dr. Mika Nagayasu, M.D., Dr. Ryuta Miyake, M.D., Dr. Masahide Nakatani, M.D., Dr. Haruki Nakamura, M.D., Dr. Shunsuke Onishi, M.D. for data collection.

## Author Contributions

**Conceptualization:** Seiji Mabuchi.

**Data curation:** Sho Matsubara, Seiji Mabuchi, Yoshinori Takeda, Naoki Kawahara.

**Formal analysis:** Sho Matsubara, Seiji Mabuchi, Yoshinori Takeda, Naoki Kawahara.

**Supervision:** Hiroshi Kobayashi.

**Writing – original draft:** Sho Matsubara, Seiji Mabuchi.

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
