## [Decision Letter · Decision Letter 0]

25 Aug 2020

PONE-D-20-05050

Prognostic value of pre-treatment systemic immune-inflammation index in patients with endometrial cancer

PLOS ONE

Dear Dr. Mabuchi,

Thank you for submitting your manuscript to PLOS ONE. After careful consideration, we feel that it has merit but does not fully meet PLOS ONE’s publication criteria as it currently stands. Therefore, we invite you to submit a revised version of the manuscript that addresses the points raised during the review process.

Minor revisions were suggested by both reviewers.  Please see the comments below.

We look forward to receiving your revised manuscript.

Kind regards,

Gayle E. Woloschak, PhD

Academic Editor

PLOS ONE

Journal Requirements:

2.We note that you have indicated that data from this study are available upon request. PLOS only allows data to be available upon request if there are legal or ethical restrictions on sharing data publicly. For information on unacceptable data access restrictions, please see http://journals.plos.org/plosone/s/data-availability#loc-unacceptable-data-access-restrictions.

3.Thank you for stating the following financial disclosure:

 [NO - Include this sentence at the end of your statement: The funders had no role in study design, data collection and analysis, decision to publish, or preparation of the manuscript.].

Additional Editor Comments (if provided):

Minor revisions were suggested by both reviewers.

Reviewers' comments:

Reviewer's Responses to Questions

**Comments to the Author**

1. Is the manuscript technically sound, and do the data support the conclusions?

Reviewer #1: Yes

Reviewer #2: Yes

2. Has the statistical analysis been performed appropriately and rigorously? 

Reviewer #1: I Don't Know

Reviewer #2: Yes

3. Have the authors made all data underlying the findings in their manuscript fully available?

Reviewer #1: Yes

Reviewer #2: Yes

4. Is the manuscript presented in an intelligible fashion and written in standard English?

Reviewer #1: Yes

Reviewer #2: Yes

5. Review Comments to the Author

Reviewer #1: Only question I'd have is the unbalanced arms. The derived systemic immune index is significantly different in for patients in different stages and grades. Would the index still be signifiicant if the arms were balanced with a propensity score matching?

Reviewer #2: This is an interesting manuscript that builds on evolving body of evidence driving toward improved indices to estimate survival after a cancer diagnosis. The index selected has been evaluated in a larger body of evidence, but not specifically for endometrial -a disease wherein survival is quite high in comparison to other cancers.

A few considerations:

Abstract -

while SII may have been determined to be an independent risk factor for survival, FIGO stage was by for the strongest predictor and this should be acknowledged in the abstract.

Overall

1. the sample was predominantly women with early disease and highly treatable endometrioid disease - yet "all patients were treated with surgery AND chemotherapy" -were women who only underwent hysterectomy excluded?

2. the mean BMI was within normal limits - this seems very unusual for endometrial cancer. The discussion should include a description and related context for interpreting the findings in countries where obesity is the major driver of disease risk. At minimum generalizability of the findings needs to be addressed more robustly.

3. Do you have data on co-morbid conditions such as diabetes? CVD? that may influence survival?

4. how readily assessed are the markers of the SII contstruct? in other words, is it reasonable to apply these over FIGO stage to estimate prognosis -or would it require additional measures be routinely assessed clinically? more clinical context to the application of these measures would be important.

6. PLOS authors have the option to publish the peer review history of their article (what does this mean?). If published, this will include your full peer review and any attached files.

Reviewer #1: No

Reviewer #2: No

---

## [Author Response · Author response to Decision Letter 0]

2 Sep 2020

Responses to Reviewer #1

Comment 1: 

Only question I'd have is the unbalanced arms. The derived systemic immune index is significantly different in for patients in different stages and grades. Would the index still be signifiicant if the arms were balanced with a propensity score matching? 

Response: 

Thank you for a reviewer’s thoughtful comment. Although SII was found to be an independent predictor for decreased PFS and OS in patients with endometrial cancer in the multivariate analysis, the prognostic significance of SII was not investigated in studies in which arms (SII-high versus SII-low) were balanced. Thus, to validate the results from current study, future studies especially in a prospective randomized or a propensity score matching settings are required. We have described this in lines 267-269 of the revised manuscript.

 

Responses to Reviewer #2

Comment 1: 

While SII may have been determined to be an independent risk factor for survival, FIGO stage was by for the strongest predictor and this should be acknowledged in the abstract.

Response: 

As suggested, we have indicated this in the Abstract.

Comment 2:

The sample was predominantly women with early disease and highly treatable endometrioid disease - yet "all patients were treated with surgery AND chemotherapy" -were women who only underwent hysterectomy excluded?

Response:

Thank you for the reviewer’s comment. We have corrected the wordings (lines 108-111 of the revised manuscript).

Comment 3:

The mean BMI was within normal limits - this seems very unusual for endometrial cancer. The discussion should include a description and related context for interpreting the findings in countries where obesity is the major driver of disease risk. At minimum generalizability of the findings needs to be addressed more robustly.

Response:

I agree with the reviewer’s comment. It has been generally accepted that high BMI is associated with increased risk of endometrial cancer development. However, BMI of the Japanese women with endometrial cancer is not so high. For example, in a study conducted by Honda T, et al, roughly 70% of Japanese endometrial cancer patients had BMI<25 (Int J Womens Health. 2012;4:207-12.), which is consistent with our study. We think the prognostic significance of SII need to be further investigated in larger studies involving patients with wide range of BMI. We have described this as a limitation of the current study in lines 258-260 of the revised manuscript.

Comment 4:

Do you have data on co-morbid conditions such as diabetes? CVD? that may influence survival?

Response: 

Thank you for a reviewer’s thoughtful comment. We did not include comorbid conditions such as a diabetes or cardiovascular diseases, which may affect patient’s survival, as a prognostic variable in our univariate and multivariate analyses. We have described this as a limitation of the current study in lines 263-265 of the revised manuscript.

Comment 5: 

How readily assessed are the markers of the SII contstruct? in other words, is it reasonable to apply these over FIGO stage to estimate prognosis -or would it require additional measures be routinely assessed clinically? more clinical context to the application of these measures would be important. 

Response:

In our multivariate analyses, the prognostic impact of FIGO stage was greater than SII (Table 3). However, whether it is reasonable to apply SII over FIGO stage or vice versa to estimate the prognosis of endometrial cancer patients remains unknown. Thus, we believe that the clinical application of SII in the survival estimation need to be further investigated in this patient population. We have described this as a limitation of the current study in lines 269-273 of the revised manuscript.

---

## [Decision Letter · Decision Letter 1]

4 Dec 2020

PONE-D-20-05050R1

Prognostic value of pre-treatment systemic immune-inflammation index in patients with endometrial cancer

PLOS ONE

Dear Dr. Mabuchi:

Thank you for submitting your manuscript to PLOS ONE. After careful consideration, we feel that it has merit but does not fully meet PLOS ONE’s publication criteria as it currently stands. Therefore, we invite you to submit a revised version of the manuscript that addresses the points raised during the review process.

Specific recommendations are listed below as minor revisions.

We look forward to receiving your revised manuscript.

Kind regards,

Gayle E. Woloschak, PhD

Academic Editor

PLOS ONE

Additional Editor Comments (if provided):

Some minor revisions are suggested.

Reviewers' comments:

Reviewer's Responses to Questions

**Comments to the Author**

1. If the authors have adequately addressed your comments raised in a previous round of review and you feel that this manuscript is now acceptable for publication, you may indicate that here to bypass the “Comments to the Author” section, enter your conflict of interest statement in the “Confidential to Editor” section, and submit your "Accept" recommendation.

Reviewer #1: All comments have been addressed

Reviewer #3: All comments have been addressed

2. Is the manuscript technically sound, and do the data support the conclusions?

Reviewer #1: (No Response)

Reviewer #3: Yes

3. Has the statistical analysis been performed appropriately and rigorously? 

Reviewer #1: (No Response)

Reviewer #3: Yes

4. Have the authors made all data underlying the findings in their manuscript fully available?

Reviewer #1: (No Response)

Reviewer #3: Yes

5. Is the manuscript presented in an intelligible fashion and written in standard English?

Reviewer #1: (No Response)

Reviewer #3: Yes

6. Review Comments to the Author

Reviewer #1: (No Response)

Reviewer #3: Addtional comments to the authors:

This is an interesting analysis of SII as potential pre-treatment predictor for survival in cohort of endometrial cancer patients.

Contrary to author’s statement as being the first study to evaluate SII in endometrial cancer, there are at least 2 published studies and these should be included in the discussion. Mirili et al (n=101) had SSI cutoff value of 1035.9 and concluded high SII, as well as PLR, NLR is associated with decreased OS. Holub et al evaluated SII, NLR, MLR and corresponding survival in endometrial cancer. Optimal cut-off was 1100.0 in this study, of which 19.4% (n=30) of patients were included. SSI was independent predictor for OS, but not for PFS or cancer specific survival.

Recent meta-analyses (Ji and Wang; Han et al) evaluating SII in gynecologic cancer also report few studies demonstrating SII correlation with decreased survival in ovarian, cervical and breast cancer, but none included endometrial cancer.

Comments:

- As previous reviewer commented, the SSI low and SSI high groups are imbalanced. Inherent limitation of a retrospective study, and the authors mention this limitation in revised manuscript. But I am concerned that this imbalance is linked to the observed statistically significant difference.

- Would be interesting to see the numbers of pts who received chemo, radiation in Table 1.

Minor comments:

- Line 103: Pre-treatment routine blood work is defined as when? Before surgery or before adjuvant therapy?

- Line 108: Clinically staged? Standard for endometrial cancer is surgical staging.

References:

Mirili, C.; Bilici, M. Inflammatory prognostic markers in endometrial carcinoma: Systemic immune-inflammation index and prognostic nutritional index. Med. Sci. Discov. 2020, 7, 351–359.

Holub K, Busato F, Gouy S, Sun R, Pautier P, Genestie C, Morice P, Leary A, Deutsch E, Haie-Meder C, Biete A, Chargari C. Analysis of Systemic Inflammatory Factors and Survival Outcomes in Endometrial Cancer Patients Staged I-III FIGO and Treated with Postoperative External Radiotherapy. J Clin Med. 2020 May 12;9(5):1441. doi: 10.3390/jcm9051441. PMID: 32408668; PMCID: PMC7291051.

Ji Y, Wang H. Prognostic prediction of systemic immune-inflammation index for patients with gynecological and breast cancers: a meta-analysis. World J Surg Oncol. 2020 Aug 7;18(1):197. doi: 10.1186/s12957-020-01974-w. PMID: 32767977; PMCID: PMC7414550.

Han X, Liu S, Yang G, Hosseinifard H, Imani S, Yang L, Maghsoudloo M, Fu S, Wen Q, Liu Q. Prognostic value of systemic hemato-immunological indices in uterine cervical cancer: A systemic review, meta-analysis, and meta-regression of observational studies. Gynecol Oncol. 2020 Oct 19:S0090-8258(20)34019-1. doi: 10.1016/j.ygyno.2020.10.011. Epub ahead of print. PMID: 33092868.

7. PLOS authors have the option to publish the peer review history of their article (what does this mean?). If published, this will include your full peer review and any attached files.

Reviewer #1: No

Reviewer #3: No

---

## [Author Response · Author response to Decision Letter 1]

8 Dec 2020

Responses to reviewers

Comment 1: Would be interesting to see the numbers of pts who received chemo, radiation in Table 1.

Response: As suggested, we have included the numbers of patients who received chemotherapy or radiotherapy in the revised Table 1.

Comment 2: Line 103: Pre-treatment routine blood work is defined as when? Before surgery or before adjuvant therapy?

Response: As stated in the title, we have investigated the “Prognostic value of pre-treatment systemic immune-inflammation index in patients with endometrial cancer”. However, as the reviewer pointed, the timing of the blood test has not been defined in the “Methods”. We now defined it in the revised manuscript (lines 115-118).

Comment 3: Line 108: Clinically staged? Standard for endometrial cancer is surgical staging. 

Response: Thank you for the reviewer’s thoughtful comment. As pointed, the standard for endometrial cancer is surgical staging. We also do surgical staging for endometrial cancer patients. However, if surgery isn't possible especially in stage IVB patients, patients were clinically staged instead. Clinical staging in such patients are allowed globally. Clinical staging has been based on the results of a physical exam, biopsy, and imaging tests. We have changed the wording regarding the staging (lines 93-94 of the revised manuscript).

Comment 4: References

Responses: Thank you for a suggestion regarding the recent researches about SII in endometrial cancer. When we submitted our manuscript to PLOS one in Feb 2020, no reports have been published. Thus, in line 75-76 of the original manuscript, we have stated “the prognostic significance of SII in endometrial cancer patients or its relative utility when compared with other inflammatory indexes has never been investigated”. Moreover, in lines 220-221 of the revised manuscript, we stated “To our knowledge, this is the first study that investigated the prognostic significance of SII in endometrial cancer patients”. However, the time has passed, and situation have been changed. As suggested, we have changed the wordings (in lines 197-198 and 206-211 of the revised manuscript), and included recent researches about SII in endometrial cancer in the “References” and a “Supplemental Table 2”.

---

## [Decision Letter · Decision Letter 2]

8 Mar 2021

Prognostic value of pre-treatment systemic immune-inflammation index in patients with endometrial cancer

PONE-D-20-05050R2

Dear Dr. Mabuchi,

We’re pleased to inform you that your manuscript has been judged scientifically suitable for publication and will be formally accepted for publication once it meets all outstanding technical requirements.

Kind regards,

Julia Robinson

Staff Editor

PLOS ONE

Additional Editor Comments (optional):

Reviewers' comments:

Reviewer's Responses to Questions

**Comments to the Author**

1. If the authors have adequately addressed your comments raised in a previous round of review and you feel that this manuscript is now acceptable for publication, you may indicate that here to bypass the “Comments to the Author” section, enter your conflict of interest statement in the “Confidential to Editor” section, and submit your "Accept" recommendation.

Reviewer #4: All comments have been addressed

2. Is the manuscript technically sound, and do the data support the conclusions?

Reviewer #4: Yes

3. Has the statistical analysis been performed appropriately and rigorously? 

Reviewer #4: Yes

4. Have the authors made all data underlying the findings in their manuscript fully available?

Reviewer #4: Yes

5. Is the manuscript presented in an intelligible fashion and written in standard English?

Reviewer #4: Yes

6. Review Comments to the Author

Reviewer #4: (No Response)

7. PLOS authors have the option to publish the peer review history of their article (what does this mean?). If published, this will include your full peer review and any attached files.

Reviewer #4: **Yes: **David Scott Miller

---

## [Editor Report · Acceptance letter]

23 Mar 2021

PONE-D-20-05050R2 

Prognostic value of pre-treatment systemic immune-inflammation index in patients with endometrial cancer 

Dear Dr. Mabuchi:

I'm pleased to inform you that your manuscript has been deemed suitable for publication in PLOS ONE. Congratulations! Your manuscript is now with our production department. 

Kind regards, 

on behalf of

Julia Robinson 

Staff Editor

PLOS ONE